# Anatomy and Neural Pathways Modulating Distinct Locomotor Behaviors in *Drosophila* Larva

**DOI:** 10.3390/biology10020090

**Published:** 2021-01-25

**Authors:** Swetha B. M. Gowda, Safa Salim, Farhan Mohammad

**Affiliations:** Division of Biological and Biomedical Sciences (BBS), College of Health & Life Sciences (CHLS), Hamad Bin Khalifa University (HBKU), Doha 34110, Qatar; SMurthyGowda@hbku.edu.qa (S.B.M.G.); sasalim@hbku.edu.qa (S.S.)

**Keywords:** brain circuits, neural communication, *Drosophila* larvae, locomotion, sensory systems, information processing

## Abstract

**Simple Summary:**

Simple movements require the involvement of many neurons before a group of muscles receives the signal to contract and relax. The sensory system adds another layer of complexity to movement generation, as the mechanism of sensory information integration is not very well understood. Despite a large body of research in this area, our understanding of organismal behavior in the context of their natural environment remains relatively poor. Observing the brain in action to be able to build neural maps and demonstrate causality is a major challenge. *Drosophila* larvae provide an excellent genetic tractable model to study behavioral response to a variety of sensory modalities and underlying neural circuitries. The understanding of genetic and physiological components of movements provides directions to understand how different locomotion types are achieved. In this review, we provide details of underlying circuitry and neural pathways required by *Drosophila* larvae for successful movements in both normal and defensive state, with an emphasis on the role of interneurons in the regulation of these movements.

**Abstract:**

The control of movements is a fundamental feature shared by all animals. At the most basic level, simple movements are generated by coordinated neural activity and muscle contraction patterns that are controlled by the central nervous system. How behavioral responses to various sensory inputs are processed and integrated by the downstream neural network to produce flexible and adaptive behaviors remains an intense area of investigation in many laboratories. Due to recent advances in experimental techniques, many fundamental neural pathways underlying animal movements have now been elucidated. For example, while the role of motor neurons in locomotion has been studied in great detail, the roles of interneurons in animal movements in both basic and noxious environments have only recently been realized. However, the genetic and transmitter identities of many of these interneurons remains unclear. In this review, we provide an overview of the underlying circuitry and neural pathways required by *Drosophila* larvae to produce successful movements. By improving our understanding of locomotor circuitry in model systems such as *Drosophila*, we will have a better understanding of how neural circuits in organisms with different bodies and brains lead to distinct locomotion types at the organism level. The understanding of genetic and physiological components of these movements types also provides directions to understand movements in higher organisms.

## 1. Introduction

Locomotion is a complex motor behavior that animals use to navigate, find food, mate, escape from life-threatening situations, and respond to different environmental sensory conditions and cues. Just a simple movement involves an intricate pattern of precise and alternating neuronal activity and inactivity, requiring continuous adjustments in appropriate muscle positions and postures [1,2,3,4]. In both vertebrates and invertebrates, locomotion is an integral component of many behaviors and many locomotion related genes are evolutionarily conserved and have human orthologs [5]. Gene or pathways affecting *Drosophila* locomotion may lead to identification of corresponding mechanisms in higher organisms including humans. 

Animals, including *Drosophila*, encounter several types of external stimuli such as olfactory, visual, temperature, and humidity sensory cues. How does the brain recognize these external stimuli with sensory receptors system, generate an internal representation of the surrounding environment, pass this information through descending and premotor neurons to Motor Neurons (MNs) for processing, and convert the sensory representation into an appropriate motor output is a topic of extensive research in many laboratories [6,7,8,9]. See detailed reviews on sensory system in [10]. Our understanding of how the central nervous system (CNS) generates these neural patterns of activity and inactivity and how the peripheral sensory system creates an internal representation of the dynamic outside world is minimal [11]. We still need to identify many of the individual neuronal components, anatomical projections, and synaptic connections that produce such complex motor behaviors. Owing to the simplicity of the nervous system and the availability of genetic tools to manipulate individual neurons, *Drosophila* constitutes a relatively accessible and genetically tractable model system to understand locomotion [12,13]. Using this model, we can delineate the underlying neuronal circuits regulating basic crawling and sensory-mediated locomotion. Importantly, *Drosophila* locomotion models are comparable, to some extent, to vertebrate models, as they share similar neuronal mechanisms that govern basic motor patterns, rhythm generation, and neuromodulation [3,14,15].

Notable advancements in neuronal and genetic aspects of locomotor control have been made possible due to the development of genetic tools to select and dissect circuits at the single neuron level [16]. By adopting these techniques, we are now obtaining an understanding of the critical genes and neurons controlling and modulating locomotion. For instance, optogenetic tools allow to either activate or inactivate any neuronal cell at a millisecond time scale [17,18]. Connectomics provides anatomical maps and allows rendering of neural circuit synaptic connectivity [19,20,21,22,23,24,25,26,27,28,29,30,31,32]. Finally, synaptic tracing methods and live imaging techniques help identify the roles of neurons in controlling different types of locomotion. 

In this review, we provide an overview of our current understanding of *Drosophila* larvae locomotion. We discuss recent advancements in understanding the anatomy and modulation of neuronal circuits that govern and control distinct locomotion types. We highlight scientific gaps and speculate on how we might resolve these at the mechanistic level.

## 2. The Topological Organization, Myotopic Map, and Neuro-Muscular Architecture of *Drosophila* Larva 

Before we decipher the mechanisms of larval movements, it is important to have a general understanding of the neuromuscular architecture of the *Drosophila* larva. In *Drosophila* larvae, the CNS is classified into brain, subesophageal ganglia (SOG), and ventral nerve cord (VNC), and in the VNC there are ~270 bilateral pairs of interneurons [33,34] that are either local or projection neurons [13]. The larval interneurons can be either excitatory or inhibitory: most excitatory interneurons are cholinergic while inhibitory interneurons are either GABAergic or glutamatergic [13]. There are also uncharacterized and poorly understood dopaminergic and serotonergic interneurons that modulate larval locomotion [35,36,37]. The scientific focus has shifted recently to identify excitatory and inhibitory interneurons controlling distinct aspects of larval locomotion [22,23,24,25,38,39]. 

The *Drosophila* larvae mainly exhibit foraging behavior throughout their lifetime to obtain sufficient nutrition required for their successful metamorphosis and transformation into adults. However, when met with a challenging situation, they also exhibit several modifications in crawling behavior, which include turns [40], head sweeps [41], bending [42] hunching [43], retreating [37], borrowing (digging) [44], escaping (rolling) [45], and cue-directed [46] and memory-guided locomotor behaviors [47]. Forward and backward movements are generated due to symmetrical muscle contractions driven by thoracic and abdominal regions that run from anterior to posterior axis of larval body. Larval behaviors such as head sweeps and turning are known to be mediated by asymmetrical muscular contractions driven by anterior thoracic segments [12,13,40] (Figure 1A). 

The *Drosophila* larva body contains three thoracic (T1-T3) and eight abdominal segments (A1–A8). Each hemisegment has ~30 body wall muscles grouped into two major sets of muscles—Longitudinal Muscles (LMs) arranged along the body axis and Transverse Muscles (TMs) arranged orthogonal to the body axis (Figure 1A) [7,8]. Thirty-six MNs innervate the hemisegment of each muscle, each projecting through Intersegmental Nerve (ISN) and Segmental Nerves (SN) to innervate internal and external muscles, respectively (Figure 1B) [48]. The internal and external muscle groups are further subdivided into ventral and dorsal groups. Each group receives input from different MN types: type 1b MNs with bigger boutons, type 1s MNs (Rp2 neuron) making smaller boutons and 1b and 1s MNs provides glutamatergic inputs to the muscles [48,49,50,51]. Each type 1b MN typically innervates one muscle and provides excitatory inputs to rhythmically active muscles during forward and backward peristalsis [13,52]. Type 1s MNs on the other hand innervate groups of muscles [53,54]. Whole-cell patch-clamp recordings of type 1b and type 1s MNs showed that type 1b MNs are recruited for making low threshold fine-scale movements, while type 1s MNs are preferred for high threshold powerful activities [55]. Also present are the neuromodulatory types II and III MNs that release octopamine to most muscles and insulin to muscle 12, respectively [26,53,56,57].

Peristaltic larval movements are generated by coordinated bilateral muscle contractions that propagate from the posterior end to the anterior end of *Drosophila* larvae, in forward locomotion. The sequential activation of MNs innervating each muscle segment results in a coordinated muscle contraction. Several studies have revealed a role for premotor interneurons in regulating this mode of larval locomotion [22,23,24,25,38,39]. However, whether MN activity is required for coordinated larval peristalsis is unclear. Larval peristaltic movements are achieved by sequential contractions of muscles; here, coordination is achieved by precise MN connectivity. MN dendritic domains form a myotopic map based on the relation between the position of dendrites and their target muscles [48]. The dorsally projecting MNs exhibit lateral dendritic arborizations while the ventrally projecting MNs have medial dendritic arborizations [13,58]. Recently, Matsunaga et al., (2017) [59], showed that MNs relay locomotion information to the neural circuitry via gap junctions to generate coordinated motor patterns during crawling in *Drosophila* larvae. The researchers used a combination of optogenetics (a method of controlling neuronal activity using light and transgenic expression of light actuators [17,18]) and calcium imaging (by genetically expressing calcium indicator like GCaMP6 [60,61,62,63]) to study the effects of manipulating MN activity in isolated larval ventral nerve cord (VNC) preparations. Optogenetic inhibition of MNs by expressing halorhodopsin (NpHR, light-gated chloride pump [64]) in specific abdominal segments (A4, A5, or A6) but not in other anterior or posterior abdominal segments resulted in a significant decrease in the frequency of motor waves. By contrast, optogenetic activation (by expressing light sensitive depolarizing channelrhodopsin, ChR2 [65,66,67]) of MNs only in posterior segments (A6 and A7) resulted in increased motor waves, suggesting a role for MNs in regulating motor wave generation [59]. 

Local photo-inhibition and photo-activation of MNs in specific segments affected the motor pattern generation and gap junction-specific inhibitors reverted the effects of the photoinhibition of MNs on motor pattern generation, these observations suggested that gap junctions regulate the information to the central circuits of the motor circuitry [59]. Indeed, gap junctions mediate MN wave frequency changes, as MN-harboring mutations in gap junction proteins ShakB2 and orge2 or cell type-specific RNAi of ShaKB2/org2 in MNs, failed to induce wave generation and rather restricted larval crawling. Finally, transmission electron microscopy (TEM) analysis has also identified intersegmental MN contacts that control motor pattern generation via gap junctions [59]. Below, we go into detail on neural circuits that regulate forward and backward locomotion with detailed roles of interneurons controlling inter and intrasegmental muscle coordination, wave propagation, coordinated symmetric motor output, speed of locomotion, and backward locomotion.

## 3. Neural Circuitry in Thorax and Abdominal Regions Controls Movements Related to Distinct Motor Patterns during Exploratory Behavior in *Drosophila* Larva

Generation of stereotypical repetitive movements is the most basic feature of complex behaviors such as breathing, walking, swimming, chewing, and crawling [3,40,41,55,68,69,70,71,72,73]. These stereotypical repetitive movements are mostly driven by central pattern generators (CPGs) [74]. The CPG-mediated motor neuronal circuitry in the CNS regulates rhythmic locomotor patterns independent of sensory inputs from the brain (Figure 1A) [2,3]. The probable existence of such CPG neuronal networks comes from studies of semi-intact preparations in vertebrate and invertebrate systems [75]. Recent electrophysiological and behavioral studies using *Drosophila* larval system have revealed that CPGs that regulate the larval stereotypical peristaltic movements are located in the thoracic and abdominal segments of VNC [40,76,77]. Stereotypical peristaltic movements generated by *Drosophila* larvae have been hypothesized to be endogenously generated in the thoracic and abdominal segments of ventral nerve cord (VNC) region. However, the nature and neuronal basis of these endogenous locomotor CPGs is yet be identified [78]. Berni et al. [40] manipulated the neuronal activity specifically in brain and SOG (labeled by BL-Gal4) and revealed that anterior segments of the nervous system (brain and SOG) are not required for normal exploratory behaviors such as straight crawls—runs and turns confirming that these exploratory behaviors are innately generated by thoracic and abdominal segments of VNC [40]. However, the inputs from the brain and SOG region were crucial for modifying the VNC generated exploratory behaviors with altered frequency and direction of turns in the presence of a sensory stimulus. Another study by SR Pulver et al. [76] used advanced imaging techniques such as calcium imaging and electrophysiological recordings and performed live imaging of isolated larval CNS to record the calcium induced motor activity in SOG, thoracic, and abdominal regions of VNC. They showed that the isolated CNS of larva generated three endogenous types of motor waves—two bilaterally symmetrical forward and backward waves that resembled forward and backward locomotion produced during intact larval crawling patterns. A third bilaterally asymmetrical motor wave pattern was generated that resembled the turning behavior of intact larvae. To determine specific regions of larval CNS that regulating distinct fictive motor patterns, motor pattern in larvae with surgically ablated brain and SOG regions was studied. While surgical ablation had no effect on the forward and backward motor wave patterns, it had an impact on the asymmetrical motor activity patterns, suggesting that the bilateral symmetrically generated forward and backward waves are intrinsic to VNC and inputs from anterior brain and SOG regions are required for producing bilateral asymmetric activity during fictive crawling. Such studies pave the way to investigate the neuronal basis of coordinated motor pattern generation regulating larval locomotion. A study from Kohaska et al. [12] identified segmental Period Positive Median Segmental Interneurons (PMSIs) in VNC and calcium imaging of these PMSIs in semi-intact larval VNC preparations showed patterns of motor activity that resembled forward and backward locomotion. These patterns of motor activity were correlated with the body wall muscle contractions during larval peristalsis. Optogenetic and behavioral analysis of PMSIs revealed that the premotor interneurons regulate the speed of larval locomotion by limiting the duration of motor bursts via inhibition [12]. Identifying such interneurons in distinct regions of nervous system will be useful further to dissect out the individual roles of each interneuron in regulation of specific locomotor behaviors such as forward and backward movements, head sweeps, turning, and rolling. 

## 4. The Neuronal Circuits That Control Forward and Backward Locomotion in *Drosophila* Larvae

*Drosophila* larvae mainly exhibit two kinds of crawling locomotion to navigate: forward locomotion and backward locomotion (Figure 2A,D). Neurologically, propagation of wave-like motor activity occurs from posterior to anterior and from anterior to posterior to cause these forward and backward movements. The neuronal circuit controlling larval peristalsis is mainly composed of various neuronal cell types such as MNs, interneurons and sensory neurons. Sequential activation of these neurons regulates intrasegmental and intersegmental muscle coordination and promotes either forward or backward locomotion. While we have an excellent understanding of MNs, their target muscles and sensory feedback neurons involved in the regulation of larval locomotion, the identity of interneurons, and how they control peristaltic locomotion is still unclear [12,13]. 

### 4.1. The Interneurons That Control Intrasegmental Muscle Coordination

In both vertebrates and invertebrates, the CPG coordinates sequential activation of MNs for rhythm generation during locomotion [38,79]. Although all MNs innervating body muscles are activated during larval locomotion, only a subset of MNs change the muscle recruitment timing for forward and backward locomotion [26]. Such differential muscle recruitment causes a wave of coordinated muscle activity that leads to coordinated muscle contractions, making a larva move from one point to another. Muscle contractions are coordinated both within a segment (intrasegment) and between the segments (intersegment). During larval forward or backward locomotion, LM’ contractions precede TM’ contractions followed by a phase of L/T (longitudinal/transverse) muscle contraction. The difference in timing of L/T muscle contraction is known as a phase delay, which is generated through a differential firing pattern from premotor neurons innervating L/T muscles [27]. 

Interestingly, a single GABAergic premotor neuron (iIN-1) labeled by R83H09-Gal4 (Table 1) that synapses specifically to four MNs (Figure 2B) innervating TMs (LT1-LT4) causes a phase delay in contraction by inhibiting MNs innervating TMs. Excitatory drive to MNs is contributed by the premotor excitatory interneurons (eIN1-3) innervating TMs only and is not involved in generating the L/T phase delay [27]. However, the functional relevance of this L/T phase delay is not known [13,32].

Coordinated muscle contraction requires coordinated activation and termination of neuronal bursting without premature MNs activation. While the neuronal mechanisms that prevent premature motor neuronal activation are still unclear, the mechanism underlying the termination of neuronal bursting has been studied in great detail. Calcium imaging studies have identified ~20 segmental inhibitory PMSIs in the ventral midline region of the larval VNC. Calcium recordings of the per-Gal4 (Table 1) labeled neurons in the segments of VNC show wave-like motor activity in each neuronal segment that propagated both in forward and backward directions. Fluorescence intensity signals in the successive segments confirm the wave-like pattern seen in PMSIs. The wave-like motor activity observed in the segmental PMSIs were also shown to be correlated with sequential body wall muscular contractions. Thus, the functional analysis of PMSIs in the segments of VNC show that the motor activity pattern of PMSIs were similar to rhythmic motor patterns of larval peristaltic movements. 

PMSIs are inhibitory glutamatergic premotor interneurons that make synaptic contacts with MNs. Activation and inhibition of PMSI activities by optogenetic stimulation revealed that they abolish larval locomotion and control the speed of locomotion by limiting the duration of motor bursts in each segment through inhibition (Figure 2C) [38]. Other sets of PMSIs (A02a and A02I) inhibit GABAergic premotor interneurons (A27j and A31k), which have inhibitory inputs to MNs that have innervations to muscles (Figure 2D). PMSIs (A02a and A02I) become activated after receiving synaptic inputs from mechanosensory neurons (bearing stretch receptors) and remove motor neuronal inhibition from MNs after their target muscle relaxes [13,22]. PMSIs labeled by per-Gal4 (Table 1) are rhythmically active in each segment and exhibit a slight delay in firing than the MNs in the same segment; they generate coordinated wave-like motor activity that moves from posterior to anterior similar to forward larval locomotion [38]. Anatomical analysis through the GFP Reconstitution Across Synaptic Partners (GRASP) technique [83] showed that the axonal projections of PMSIs (A02b and A02m) neurons are confined to the same segment, and they are direct local interneurons (Figure 2D).

Another class of glutamatergic inhibitory interneurons, known as glutamate ventrolateral interneurons (GVLIs), synapse to muscle MNs (Figure 2D) and are rhythmically active during larval locomotion [84]. However, GVLIs are active at different phases of the locomotor cycle than PMSIs, which are activated later. While PMSIs control the duration of motor bursts and the speed of locomotion, GVLIs influence the termination of motor bursting, being active at later phases of the locomotor cycle.

### 4.2. Intersegmental Feedback Neurons Regulate Bidirectional Larval Movements

Each larval abdominal hemisegment consists of two sets of antagonistic muscles: LMs-controlling longitudinal contractions and TMs-regulating circumferential contractions of muscles. These muscles are active during forward and backward locomotion and are sequentially coordinated in each abdominal segment, with LMs being active first [68].

Antagonistic muscles are coordinated during intersegmental motor coordination and the underlying neuronal network regulating muscle coordination is unknown. Using serial section TEM (ssTEM) reconstruction mapping and calcium imaging, researchers have identified two bilateral pairs of premotor excitatory interneurons in each segment that provide delayed intersegmental feedback to adjacent anterior or posterior neuromeres regulating bilateral motor coordination [22]. These two identified pairs of high-order feedback interneurons, namely, Ifb-Fwd (intersegmental feedback-forward cholinergic neuron (Table 1) and Ifb-Bwd (intersegmental feedback-backward neuron (Table 1) ultimately promote bidirectional motor wave propagation [22,38]. Ifb-Fwd neuronal activity causes contraction of TMs, while Ifb-Bwd neuronal activity inhibits LMs (Figure 2E). Calcium imaging further revealed that Ifb-Fwd and Ifb-Bwd neurons are distinctly active during forward and backward crawling, respectively, suggesting that these feedback neurons oppose bidirectional larval movements. Reconstruction mapping of ssTEM images identified A01c and A03g as cholinergic excitatory premotor interneurons that innervate and excite transverse MNs and their innervating muscles (Figure 2E). A02e, A02g, and A03g are PMSIs that provide inhibitory inputs to longitudinal MNs and their targeting muscles (Figure 2E). Reconstruction of ssTEM images further identified that Ifb-Fwd and Ifb-Bwd are the only typical upstream neurons to A02e and A02g PMSIs [22]. 

The EM-reconstructed circuit module explains a mechanism for sequential activation of a group of TMs and simultaneous inhibition of an antagonistic group of LMs that are required for intersegmental motor coordination [22]. Inhibiting either Ifb-Fwd or Ifb-Bwd neuronal activity by expressing inward rectifying potassium channels (Kir) resulted in reduced muscle contraction. This finding suggests that these intersegmental feedback neurons facilitate coordinated TM contraction. Connectivity data from Ifb-Fwd and Ifb-Bwd neurons and their shared target excitatory (A01c and A03g) and inhibitory (A02e, A02g and A03g) interneurons revealed that they converge at a common neuropil centrolateral region that differs from other functional motor or sensory domains. 

Previously identified inhibitory and excitatory interneurons also help regulate wave propagation and the speed of locomotion [1,2,3,4,20,22,25,38]. However, mechanisms that lead to CPG activation and that regulate rhythm generation are still not understood. Identifying the upstream and downstream circuitry of these intersegmental feedback neurons (Ifb-Fwd and Ifb-Bwd) is needed to determine these CPG mechanisms.

### 4.3. The Interneurons That Regulate Feedforward Wave Propagation

Coordinated muscle contractions within segments and coordinated muscular activity between segments allows active wave propagation during forward locomotion. Muscles in each segment sequentially contract and relax to move forward or backward and contraction in one segment is followed by relaxation of the immediate anterior segment required for successful wave propagation. Each segment is innervated by a chain of excitatory and inhibitory interneurons that promote feedforward wave propagation. Advanced high-resolution calcium imaging has identified neuronal circuits consisting of excitatory and inhibitory interneurons promoting feedforward wave propagation. These neuronal circuits consist of alternating inhibitory interneurons, GABAergic dorsolateral (GDL) inhibitory interneurons, and a cholinergic premotor excitatory interneuron A27h (Figure 3A). The GDL interneurons are a pair of inhibitory interneurons that can be selected in larvae using the 9-20-GAL4; iav-GAL80 line [69]. A combined line 9-20-GAL4; iav-GAL80 (Table 1) allows for selection of only GDL interneurons by blocking Gal4 expression in mechanosensory neurons [25] (Figure 3B). GDL interneurons show wave-like motor activity that occurs before MNs; blocking their synaptic transmission using tetanus toxin light chain resulted in abnormal locomotion [25]. A TEM reconstruction revealed that A27h, excitatory premotor interneurons labeled by R36G02-GAL4 (Table 1) that are active only during forward locomotion, act downstream of GDL interneurons and receive inhibitory inputs from GDL interneurons in the same segment; they simultaneously provide inputs to GDL interneurons in the next segment such that there is feed-forward wave propagation [25]. Thus, A27h act as excitatory interneurons to activate MNs in one segment and activate GDL interneurons in neighboring segments to inhibit the A27hs and promote modeled feed-forward circuitry (Figure 3C).

GDL interneurons also receive proprioceptive inputs from Vpda (stretch receptors) class1 dendritic neurons [85,86] that function as modulatory sensory neurons (Figure 3C). Other somatosensory neurons such as ventral dendritic arborization VdaA/VdaC, labeled by tutl-GAL4 (Table 1) and class II dendritic arborization neurons are known to be synaptically connected to both GDLand A27h neurons (Figure 3C). However, the functional role of these sensory neurons remains unknown.

The main downstream target of GDL interneurons is A27h premotor interneuron. GDL interneurons provide synaptic inputs to glutamatergic PMSI A02d that controls locomotion speed [25,38]. The A02d interneurons, along with and GDL interneurons, integrate inputs from somatosensory neurons and regulate proprioceptive signals within its segments, indicating the possible role of GDL interneurons in controlling the speed of locomotion and posture (discussed in the proprioception section in detail). Additional sets of excitatory and inhibitory interneurons have been identified, for example, glutamatergic interneurons; however, their role in regulating wave propagation is unknown [38]. A descending neuron located in the SubEsophageal Zone (SEZ) that synapses with an A27h excitatory interneuron was also found to control wave propagation [25]. GDL interneuron also synapses with several other premotor interneurons, whose role in the wave propagation has not been experimentally determined for their role in regulating backward locomotion [68]. 

### 4.4. Moon Descending Neurons Activate Backward Locomotion

As described previously, A27h excitatory premotor neurons are active predominantly during forward locomotion, implicating the possibility of a parallel neural circuit that regulates backward locomotion. Neuronal activation of segmentally repetitive VNC-located “wave neurons” projecting to anterior body segments induces backward locomotion [24]. In addition, these wave neurons integrate sensory inputs from multidendritic class IV neurons (described in Section 2) and have a role in nociception (discussed in the wave neurons mediated pathway section). 

The role of descending neurons that control backward locomotion and mechanisms that regulate backward peristalsis and the collective suppression of forward locomotion is largely unknown (Figure 3D). However, researchers have identified a unique set of descending neurons controlling backward locomotion, termed Moon Descending Neurons (MDNs) labeled by R49F02-AD; R53F07-DBD (Table 1). These bilateral pairs of command-like descending neurons in larvae (Figure 3E) are coordinated to activate backward locomotion and simultaneously deactivate forward locomotion [23]. An EM-reconstruction analysis identified MDNs as presynaptic to excitatory cholinergic premotor neurons (A18b) labeled by R94E10-Gal4 (Table 1) that promote only backward locomotion (Figure 3F). Through optogenetic activation assays and calcium imaging studies, it was further revealed that these MDNs activate GABAergic inhibitory (Pair1) neurons labeled by R75C02-LexA (Table 1) that drive inhibition to A27h premotor interneurons (forward motor wave promoting neurons) and help to suppress forward locomotion (Figure 3F) [23,25]. Circuit mapping data of MDNs also indicated that forward and backward locomotor behaviors behave antagonistically by receiving excitatory and inhibitory inputs from descending neurons. Using split-Gal4 intersectional genetic systems, these larval MDNs were shown to persist through adult stages and promote backward locomotion [87]. 

It thus seems that MDNs control the switch from forward to backward locomotion. Whether similar command-like descending neurons exist for controlling the opposite switch of locomotion from backward to forward, is unknown. MDNs mainly send their projections to thoracic regions and anterior abdominal regions (A3–A5), thus regulating backward locomotion; perhaps descending neurons controlling forward locomotion have projections extending into posterior abdominal segments. Functional characterization of descending neurons controlling forward locomotion would help identify the inhibitory synaptic connections to Pair1 neurons and A18b neurons that activate backward locomotion.

### 4.5. Even-Skipped Lateral Interneurons Regulate Bilateral Motor Coordination for Symmetric Motor Output

To achieve coordinated peristaltic locomotion, a symmetrical motor output is required. Precise muscular contractions mediate bilateral symmetrical motor output. These contractions are achieved through intrasegmental and intersegmental motor activity. The synchronous muscle contractions between the left and right sides of the body are also required to generate symmetrical motor output [82]. The neural circuitry controlling the bilateral symmetrical motor output is mostly unknown [82]. One recent neurogenetic screen identified a set of interneurons known as Even-skipped lateral (EL) interneurons that express the evolutionarily conserved transcription factor Even-skipped (Eve) [88]. These ELs are sensory processing interneurons present in the nervous system of all bilaterally symmetrical animals [82,88,89]. In *Drosophila* larva, Eve is expressed in the subsets of interneurons and MNs in each body segment, but not in the brain [39,82,90]. Further, neuronal stem cell lineage analysis combined with intersection genetics, optogenetics, connectomics, calcium imaging, and behavior monitoring identified that the early-born ELs are part of a mechanosensitive-escape circuit while the late-born ELs mediate a circuit involved in a proprioceptive body posture [91].

Researchers showed that activating all EL interneurons labeled by EL-Gal4 (Table 1) via thermal and optogenetic activation techniques resulted in the first transient increase followed by sustained slower crawling speeds along with abnormal left-right body asymmetry (C-bends) [91]. Similarly, inactivation of all EL neurons resulted in a slow locomotion speed and characteristic “wavy body postures” and asymmetric muscle contraction amplitudes [39,82,90]. Interestingly, selective optogenetic activation of early-born ELs labeled by EL-Gal4; R11F02-Gal80 resulted in a transient increase in larva crawling speed but no sustained reduction in speed. By contrast, selective activation of late-born ELs labeled by R11F02-Gal4 (Table 1) resulted in a decrease in larval speed with abnormal left-right body posture [91].

EL interneurons comprise a sensorimotor circuit that receives direct input from several proprioceptive neurons (Vbd, dbd, and lesA) through Jaam 1–3 interneurons; they are also connected to motor output neurons through 2-Saaghi interneurons (Figure 4A–C) [39,82,90]. Although the function of these proprioceptive sensory neurons (Vbd, dbd, lesA) is unclear, they might deliver muscle contraction amplitude information to EL interneurons. While early and late-born ELs are part of different sensory circuits, EL interneurons are part of a sensorimotor circuit that modulates left-right symmetric motor outputs and are not part of CPGs, as the timing of muscle contraction is unaffected during larval locomotion.

### 4.6. Thoracic Neuronal Network Regulates Turning and Head-Sweeps 

Apart from forward and backward locomotion that requires symmetric control on movements, *Drosophila* larvae also make turning and head-sweeps behaviors (Figure 1A). Turning and head sweep behavior might be regulated by asymmetric or unilateral episodic motor activity, which is confined to thorax and anterior abdominal segments [77], however specific neurons or neuronal network mediating these asymmetric or unilateral motor activity have not been identified. To identify specialized thoracic regions in the larval brain that are controlling turning or headsweep behaviors, Jimena Berni took the genetic approach and studied the axonal guidance pathway of slit-robo mutants with defects in the neuronal midline crossing. Genetic and behavioral analysis of robo mutants and different allelic combinations of robo mutants showed abnormal exploratory behavior with circular paths and significant reduction in pause turns as compared to control larvae with straight paths and turns [77]. These robo mutants with axonal midline crossing defects could generate normal coordinated crawling peristalsis, indicating that bilateral symmetrical contractions normally mediate wave propagation functioning despite abnormal midline connectivity. Besides significant decreased turns displayed by robo mutants, there was significant increase of head rearing behavior (head-cast) generated due to sequential bilateral symmetrical contractions of thoracic segments [77]. Through calcium imaging experiments it was determined that CPG exploration circuitry was normal and not involved in the abnormal behavioral phenotypes of robo mutants, as bilateral symmetrical waves of motor activity were detected along the thoracic and abdominal segments in the semi-intact larval preparations that resembled the forward and backward locomotion in intact larval crawling. Additionally, asymmetric unilateral patterns of motor activity in anterior segments were observed, which were similar to asymmetric motor activity regulated turning behavior. These asymmetric activity-induced turns were completely abolished in robo mutants, which show that the axonal midline connectivity is crucial for the generation of turns. Further, to dissect the more specific regions of thoracic and abdominal regions responsible for generation of asymmetric motor outputs (turn), commissureless (regulator of robo), with similar axonal midline defects as robo, was used and specifically targeted to thoracic and abdominal regions alone. It was shown that the asymmetric motor outputs induced turns are unique to thoracic regions and propagated further down to abdominal segments and thus confirm that thoracic regions are responsible for generating asymmetric unilateral motor activity associated turning behavior [77]. 

### 4.7. Distinct PMSIs Regulate Specific Locomotion Types

PMSIs regulate larval locomotion speed by inhibiting the duration of the MN output (Figure 1D) [38]. A subset of these PMSIs in larval ventral nerve cord express the Maf transcription factor, Traffic jam (tj), and their role in regulating locomotion speed in *Drosophila* has been characterized [39]. However, unlike PMSIs, tj+ interneurons can be glutamatergic, cholinergic, or GABAergic: five tj+ interneurons per hemisegment are glutamatergic, 10 tj+ interneurons are cholinergic and eight tj+ interneurons per hemisegment are GABAergic [39]. Activating tj+ brain expressing neurons has no defects on larval crawling speed, indicating that tj+ -VNC neurons are involved in inducing paralysis effects in larvae. Activating specific subsets of tj+ neurons based on their neurotransmitter identity also influences specific locomotor behaviors. For example, activating cholinergic tj+ interneurons result in abnormal crawling behaviors with crescent shape muscle contractions, rolling behavior, and decreased peristaltic waves. Activity in glutamatergic tj+ also induces paralysis along with rapid headcast, and activation of tj+ GABAergic interneurons’ activity displayed paralysis and reduced crawling speed in larvae [39]. 

Further characterization of tj+ GABAergic midline interneurons revealed that they also co-expressed period (per): ultimately, three per+/tj+ GABAergic midline interneurons were found. Triple intersectional labeling and activation showed that these three per+/tj+ GABAergic neurons affect crawling speed and derive from median neuroblast (MNB) progeny. These per+/tj+ GABAergic MNB neurons are GABAergic Ladder neurons-feed-forward inhibitory interneurons that control mechanoresponsive choice behavior [28] and modulate the speed of larval locomotion. Overall, it seems that Maf-expressing tj+ GABAergic interneurons control normal locomotion in *Drosophila* larvae. In addition, evidence suggests that mechanosensory input can activate tj+ GABAergic interneurons that inhibit Basin neurons and modulate motor output either through brain projections or directly through projections with premotor neurons [29].

## 5. Sensory Neurons Mediate Defensive Behaviors

All organisms rely on sensory neurons to detect and avoid noxious stimuli. In response to light or touch stimulation, *Drosophila* larvae exhibit turning behavior—a combination of rearing, bending, and retreating as well as various aspects of escape behaviors, such as sensing the vertical environment and turning away from noxious stimuli [37,92,93]. In response to harmful sensory stimuli, larvae exhibit a stereotyped defensive rolling behavior, where larvae roll up their body in corkscrew-like fashion [45,59,94,95]. When stimulated by a strong mechanical force, high temperature or when under predatory attack from wasps (*Leptopilina boulardi*), *Drosophila* larvae exhibit C-shaped body bending and lateral turning (also known as rolling behavior) as a defense mechanism. Intriguingly, larvae typically move towards rather than away from stimuli. The reason is unclear, but one study shows that when under wasp attack, the larva wraps around the wasp’s sting and flips it through the air and onto its back, which helps the larva to escape [94].

In *Drosophila* larvae, two morphologically distinct somatosensory neurons: type I and II neurons facilitate stimulus detection. Type I neurons are simple and associated with chordotonal sensory organs; they respond to simple light touch stimuli. In contrast, type II neurons are multidendritic and respond to diverse noxious, thermal or mechanical stimuli [96]. Type II neurons are classified further into four different multidendritic neuron classes, distinguished by their dendritic arborization patterns, indicating their diverse function [97]. Classes I–II are relatively simple, while III–IV are more complex [97]. 

The multidendritic class IV neurons (MD-IV) are multimodal nociceptive neurons that project to the entire epidermis covering the larval body required for generating rhythmic larval locomotion. Silencing neuronal activity from peripheral and MD sensory neurons leads to decreased locomotion speed and immobility; thus, inputs from the sensory feedback system are essential for modulating the generated motor outputs [98]. MD-IV neurons respond to noxious stimuli such as heat, light, and mechanical force [45,93,94,99,100,101].

Optogenetic activation of MD-IV neurons in larvae induces rolling behavior, while silencing of these sensory neurons abolishes thermally induced rolling behavior. Thus, this class of multidendritic neurons is sufficient and necessary for generating rolling-like escape behavior [94]. These nociceptive neurons also control several other sensory system-mediated escape behaviors, including pausing, bending, and hunching [23,29,94,102]. Mechanistically MD-IV neurons express the Degenerin/Epithelial Sodium Channel (PPK), which is required for mechanical nociception. As such, PPK mutants show defective responses to mechanical stimuli [45,99,103]. 

Apart from MD-IV neurons, the serotonergic neurons in the larva VNC suppress the turning behaviors in *Drosophila* larvae. These serotonergic neurons innervate to 5HT1B-receptor-bearing and neuropeptide leukokinin releasing neurons [37]. Interestingly, by expressing the temperature-sensitive dynamin protein, shibire (*Shit^s^*) in 5-HT1B-receptor cells in VNC both rearing behavior and turning behavior were blocked.

### 5.1. Mechanical Nociception

*Drosophila* larvae use tactile information to forage and escape [43]. *Drosophila* larvae exhibit multiple behaviors in a response to various tactical stimuli, and few neural pathways have been identified; we describe some of these pathways as having a role in mechanical nociception.

#### 5.1.1. Chordotonal Organ Mediated Pathway

In response to an air puff—a gentle mechanical stimulus—*Drosophila* larvae perform behavioral sequences involving bending and hunching during exploratory behaviors (Figure 5A,B). The circuitry regulating particular behavioral choices and transitions between the two behaviors are not fully known. However, one study [30] found that the mechanosensory chordotonal (mechano-ch) neurons control air puff-induced hunching and bending behavior in *Drosophila* larvae (Figure 5A,B) [30]. 

Jovanic et al. [30] showed that somatosensory projection neurons (Basin-1 and Basin-2, have basin-shaped arborizations) that are downstream to mechanosensory-chordotonal neurons, promote either bending or hunching behavior. The Basin neurons are segmentally repeated projection neurons located in the ventral sensory domain of the nerve cord. Activation of the Basin-1 labeled by R20B01-Gal4 (Table 2) neuron promotes hunch behavior and coactivation of all Basin neurons labeled by JRC-R72F11-GAL4 promotes bending by suppressing hunching. Based on synaptic connectivity to Basin neurons, Jovanic et al. characterized two types of local interneurons further; Feedforward-iLNs and Feedback-fbLNs that receive inputs from Basin projection neurons [30]. Based on the number of synaptic connections made to Basin neurons, they could divide Feedforward iLNs into Feedforward iLNa and iLNb types. They also reported two types of feedback neurons: fbLN-Ha and fbLN-Hb. FbLNs are GABAergic while iLNs are either cholinergic or GABAergic and reciprocally inhibit each other [13,30]. The Feedforward iLNa-Griddle 2 neurons driven by the JRC-SS0918-split-Gal4 line (Figure 5A,B) promote hunch behavior and mediate behavioral choice via disinhibition. The GABAergic Feedback fbLN-Hb neuron driven by JRC- SS0888-split-Gal4 line (Table 1) controls the reversal of behavioral sequences via disinhibition. Finally, lateral disinhibition of fbLN-Ha neurons promotes transitions between two behavioral sequences [30].

#### 5.1.2. Wave Neurons Mediated Pathway

In response to a noxious mechanical force, *Drosophila* larvae exhibit escape behavior depending on the segment-specific noxious mechanical stimuli they receive [24,108]. A larva moves backward when touched either gently or noxiously on its head (anterior end), and it moves forward when touched on its tail (posterior end) (Figure 5C,D) [24,43,108]. Wave neurons have been implicated in this segment-specific induced escape behavior (Figure 5C) [21]. These neurons are homologous, command-like somatosensory interneurons (A02o), present in each abdominal muscle segment [24]. Optogenetic activation of A02o wave neurons labeled by VT25803-Gal4 (Table 1) in anterior and posterior abdominal segments, elicits backward and forward locomotion, respectively (Figure 5C,D) [24]. 

Wave neurons receive synaptic input from both nociceptive multidendritic (MD-IV) and mechanoreceptive (MD-III) sensory neurons. Inhibiting the neurotransmission of these sensory neurons results in a reduced response on backward locomotion, indicating that mechanosensory neurons partially regulate these touch response behaviors. The A03a5 premotor interneurons are postsynaptic to the Wave neurons responsible for backward touch responses; they regulate other types of escape behaviors, such as rolling and bending (Figure 5C,D) [24]. 

#### 5.1.3. Basins-Goro Neurons Mediated Pathway

Using electron microscopy reconstruction, behavioral and physiological studies Ohyama et al. [56] have identified a complex multisensory circuit that integrates multimodal sensory inputs and initiates the selection of the fastest mode of escape (rolling) behavior in *Drosophila* larvae. Combined activation of nociceptive MD-IV-multidendritic neurons, as labeled by Md-IV-Gal4 (Table 2), and mechanosensory stimualtion (chordotonal neurons labeled by ch-Gal4 and iav-Gal4) (Table 2), induced enhanced escape-like rolling behavior [24,29]. These chordotonal and nociceptive neurons further converge on to first order multisensory projection neurons termed Basin neurons labeled by R72F11-Gal4 (Table 1) (named after their basin shaped arbors). Classified distinct types of these Basin neurons (Basin1­–Basin4) integrated a combination of chordotonal and MD-IV nociceptive inputs and increased the selection of the rolling behavior. Specifically, selective activation and inactivation of Basin-4 neurons triggered selection of rolling behavior and a significant decrease in rolling, respectively. These multisensory interneurons (Basin neurons) further converge into downstream neurons command-like Goro neurons (pair of thoracic neurons) labeled by R69F06-Gal4 (Table 2) and thermogenetic activation of Goro neurons also enhanced the selection of rolling behavior (Figure 5E,F) [24,29]. Activation of these Basin neurons elicited calcium levels in Goro neurons, indicating that they are functionally connected and display similar rolling behavior. 

Furthermore, the anatomical and functional connections between Basin and Goro neurons and whether integrated multimodal sensory information received by Basin neurons is converged to downstream Goro neurons were confirmed by EM volumetric analysis, which revealed two different Basin neuron mediated pathways that involve either the VNC or the brain. In the VNC pathway, the Basin neurons project to Goro neurons and Goro neurons project to the A05q and A23g interneurons in the VNC. A Basin-Goro VNC pathway thus together constitutes a local multisensory and unisensory input. In the brain pathway, the Basins indirectly synapse to Goro neurons through A00c ascending neurons. These neurons project to the brain to make up the Basin-Goro brain pathway. This pathway controls global body-wide information from both nociceptive and mechanosensory neurons (Figure 5E). Thus, chordotonal and nociceptive sensory neurons converge to distinct types of multisensory Basin neurons that are the first level of multimodal convergence and information from distinct types of Basin neurons, which converge further to downstream Goro neurons which deliver second level multimodal integration. Together, the key findings from this study indicate that there exists a multimodal integration at different levels that induces enhanced selection of rolling behavior in *Drosophila* larvae. 

#### 5.1.4. DP-ilp-sNPF Receptor Mediated Pathway

The intensity of *Drosophila* larvae behavioral response depends on the strength of the noxious stimulus. In response to a strong stimulus, the *Drosophila* larva crawls using its fastest mode of escape behavior. Crawling in this rapid manner involves two modes of locomotion, namely rolling followed by fast crawling. While fast crawling is evoked in response to mild noxious and strong mechanosensory cues, rolling is evoked only upon the strong noxious stimuli that integrate both nociceptive and mechanosensory cues [24,29]. A strong noxious mechanical stimulus may activate class II da (C2da), class III da (C3da), and Md-IV (C4da) sensory neurons, which in turn activate Dorsal pair-insulin like peptide (Dp-ilp7) neuropeptide neurons [107]. The activated Dp-ilp7 neuron releases sNPF required for paracraine feedback via sNPF-R to Class II/IIIda and MD-IV da sensory neurons. sNPF-R signaling in turn activates Md-IV(C4da) neurons that result in calcium responses in A08n neurons (labeled by R82E12-Gal4) and induce mechano-mediated nociceptive rolling behavior (Figure 5F) [107]. 

The C2da and C3da sensory neurons also express Ripped Pocket (*rpk*), No Mechanoreceptor Potential C (*nompC*) and NMDA Receptors 1 and 2 (Nmdars). These genes have critical roles in mediating behavioral responses to touch. In particular, rpk is expressed in all multidendritic neurons and is essential for the gentle touch response [109]. Md-IV neurons also express several channels that are sensitive to mechanical touch, including pickpocket (*ppk*), *ppk26* (*balboa*), *piezo*, mechanosensitive *TRP* channel *NompC* and thermo and mechanosensitive TrpA1 [45,85,99,110,111,112]. Interestingly, the *ppk26*-expressing C4da neurons mediate mechano-nociception but have no effect on thermal-nociception (Figure 5F) [110].

### 5.2. Thermo-Nociception

In *Drosophila* larvae, strong noxious heat (>39 °C) stimuli evoke corkscrew-like rolling behavior with body bending. A set of interneurons termed Down and Back neurons (DNB) are identified to respond to heat stimuli and evoke nociceptive rolling and C-shaped body bending behaviors [77]. Single-cell analysis of DNBs revealed that these neurons project downwards and send their arborizations back to their cell body. For this reason, they gained their name as ‘Down and back’ neurons (A091). Morphologically, DNBs, labeled by the 412-Gal4 (Table 2), are segmental interneurons that project to the ventromedial neuropil region. They receive inputs primarily from MD-IV sensory neurons and other Class II and Class-III (mild-touch) sensory neurons (Figure 6) [31]. 

Nociceptive integrator and premotor neurons have the highest number of connections with DNBs. The nociceptive integrator neurons are TePn05, receiving inputs from multiple nociceptive neuron types. TePn05 are postsynaptic to both DNBs and Md-IV neurons and presynaptic to both Basin-2 and Basin-4 nociceptive interneurons and Wave neurons [31] (Figure 6). DNBs are downstream to MD-IV nociceptive neurons that project to premotor PMSIs-A02e and A02g and are indirectly linked to Goro command neurons (Figure 6) [31]. Inactivation of Goro neurons with DNBs’ activation abolishes rolling and promotes C-shaped body bending, suggesting that DNBs control C-shaped body bending behavior in *Drosophila* larvae [31].

MD-IV neurons express TRP channels, which are homologous to vertebrate temperature-sensitive TRPA1 channels in nociceptive neurons [45]. *Drosophila* larvae TRPA1 channels (e.g., painless) are unable to respond to heat [45,99]. The MD-IV neurons also bind pre-synaptically with segmentally repetitive medial clusters of C4 da second-order interneurons mCSIS, labeled by R94B10-Gal4 (Table 2), which synapse to SNa BarH1 labeled by BarH1-Gal4 (Table 2) as motor output neurons [105] (Figure 6).

Recently, Dason et al. described the role of neurons expressing a foraging gene (*for*) that regulates thermal nociception in *Drosophila* larvae [106]. *Drosophila* encodes a guanosine 3’, 5’ cyclic monophosphate-dependent protein kinase (*PKG*) and carries four promoter regions (pr1-4); of note, pr1-Gal4 expression was detected in the larval CNS. Optogenetic activation of pr1-Gal4 (Table 2) labeled pr1 cells induced curling and rolling behavior, revealing the role of pr1 cells in thermal nociception. Two allelic variants of for, known as rover and sitter, were further examined for their response to thermal nociception and were found to have a significantly shorter response latency (i.e., the time in seconds for the onset of rolling behavior of rovers compared to sitters) [106]. Expression analysis showed that rovers had higher pr1 mRNA levels compared to sitters, indicating that rovers are more sensitive for thermal nociception. The researchers also generated null mutants and found that they had longer response latencies than rovers and sitters. Expression of *for* in pr1 cells in the *for* null mutant background shortened the response latency period. This genetic manipulation therefore revealed that pr1 cells are required for thermal mediated rolling behavior. 

pr1-Gal4 is expressed in tracheal multidendritic neurons on the larval body wall but not in MD-IV multidendritic neurons. pr1-Gal4 is also expressed in VNC neurons and synapses to MD-IV multidendritic neurons. Finally, optogenetic activation of pr1 cells at different stages of larval development induced curling and rolling behavior. It thus seems that these neurons in the sensory circuit mediate thermal-induced rolling behavior developmentally [106]. 

### 5.3. Chemotaxis

For insects, olfaction serves as a primary sensory system. For example, *Drosophila* larvae exhibit robust orientation behavior towards attractive food odors and avoidance of sites associated with parasitoid smells [113,114]. The *Drosophila* larval olfactory response (chemotaxis) results from coordinated transitions between four different behavioral states: runs (forward motion), stops (termination of ongoing peristaltic waves), heads casts (lateral head sweeps), and turns [46,115].

Researchers have identified that an olfactory descending neuron (PDM-DN) labeled by PDM-DN Gal4 (Table 1) mediate the sensory-motor pathway in larval chemotaxis to promote run-to-turn transitions [32]. Optogenetic activation of the PDM-DN neuron initiates stopping behavior by blocking forward propagating waves and induces turning behavior by favoring head sweeps (Figure 7A) [32]. Anatomically, PDM-DN neuronal bodies are located in the posterior-dorsal medial regions of the brain. Their dendrites arborize into the mushroom body (MB) and lateral horn (LH), indicating that PDM-DN neuron receive olfactory inputs. EM analysis of PDM-DN revealed that a cholinergic excitatory premotor interneuron A27h [25] is synaptically connected to PDM-DN via the GABAergic downstream neuron SEZ-DN1 (also known as Pair1, labelled by R75C02 (Table 1)) and has an inhibitory effect on A27h; it thus promotes stopping behavior and controls run-to-turn transitions similar to PDM-DN (Figure 7B) [32].

The larval response to the intensity of odor concentrations is mainly processed in the antennal lobe. However, the role of different brain regions in regulating larval chemotaxis is unclear [80]. A set of excitatory neurons, known as Odd neurons (expressing odd-skipped transcription factors), are involved in larval chemotaxis (Figure 7). Odd neurons labeled by Odd-Gal4 (Table 1) function downstream of antennal lobe centers: they project their dendrites to the MB-calyx region and axons to the central brain [80] and receive synaptic inputs from projection neurons in the MB calyx. Calcium imaging of odd neurons revealed that larvae respond to attractive odors such as Apple Cider Vinegar. Selective silencing of odd neurons resulted in abnormal chemotaxis and increasing their excitability increased the larva’ sensitivity to odor detection [80]. It is worth noting that manipulating the Odd-neurons function affects chemotaxis at lower (1:20 dilutions) but not higher odor concentrations. As silencing of Odd-neurons does not abolish chemotaxis, this suggests the existence of a parallel neural pathway involved in chemotaxis [80].

The sensory motor pathway regulating larval chemotaxis has also been mapped and through genetic and behavioral screening a set of neurons located in the SEZ have been identified that control reorientation behaviors during chemotaxis (in response to an odor source 1-hexanol) [81]. Specifically, a loss-of-function genetic screen identified NP4820-labeled neurons in larval CNS; blocking the neurotransmission of these neurons by expressing tetanus toxin light chain resulted in abnormal chemotactic phenotypes with frequent halts and reversals as compared to controls. Detailed analyses of NP4280 neurons revealed that NP4280-labeled neurons are involved in transitions from run-to-turn behavior [81]. Both optogenetic and thermogenetic activation of NP4820-labeled SEZ neurons (Table 1) resulted in the enhanced reorientation behaviors, supporting a role for the SEZ in chemotaxis. Further, silencing a subset of SEZ neurons using R23F01-Gal4 (Table 1) targeting four to six neurons in SEZ led to a similar loss-of-function phenotype of NP4280 neurons; optogenetic activation of R23F01-targeted SEZ neurons was sufficient to induce reorientation behaviors [81]. Together, these data implicate a role for SEZ neurons in integrating sensory inputs and generating sequential motor sequences.

### 5.4. Chemo-Nociception

The olfactory circuits involved in the chemotactic response are known, but the larval response to a nociceptive or tissue-damaging chemical or to an environmental irritant is unclear. Researchers have, however, developed a novel behavioral assay to study chemically induced nociceptive behavior in *Drosophila* larvae [96]. In this assay, larvae are briefly exposed to the noxious chemical stimulus HCl (hydrochloric acid): exposure to low concentrations of HCl (0.5%) does not evoke a nociceptive response, while exposure to high concentrations (1­–9%) evokes aversive rolling-like behavior. Of note, the larval responses to other chemicals, such as sulfuric or acetic acid, are similar to HCl. Using this approach, Im et al. found that exposure to a noxious acid stimulus causes a decrease in the survival rate and also tissue damage in the third instar larvae. The cellular stress response to HCl was induced by activation of msN-lacZ, a reporter of the JNK signaling pathway [96]. Furthermore, through inactivation strategy (expressing active tetanus toxin in multiple classes of multidendritic sensory neurons, it was found that Classes IV, III, and I neurons mediated the chemical nociceptive response in stimulated larvae. 

The researchers saw an increase in calcium activity, as measured by Calcium Modulated Photoactivatable Ratiometric Integrator (CaMPARI) [116] upon HCl activation of Md-IV neurons, revealing the primary role of these neurons in chemical nociception [96]. They also elucidated that Basin-4 interneurons labeled by JRC-SS00740-Gal4 (Table 1) acted as downstream targets of Class IV neurons to mediate chemical nociception [96]. Finally, sensitization of chemical nociceptive responses by physical puncture wounding resulted in increased nociceptive response to subthreshold HCl concentrations [96].

### 5.5. Proprioception or Kinaesthesia

Similar to other animals, *Drosophila* larvae can sense the motion or position of their body parts via a process known as proprioception [117]. Dedicated proprioception neurons regulate coordinated movements by providing feedback of static and dynamic positions of body parts to the CNS that are critical to coordinating movements. Proprioception is helpful in adjusting body movements in changing and unpredictable situations such as defense, escape, navigation, and mating. In *Drosophila*, the neural mechanisms regulating proprioception are not fully understood. However, specialized sensory chordotonal (Cho) neurons [118], multidendritic neurons [119], and bipolar dendritic neurons [69,120] have been implicated in proprioception-mediated movements in *Drosophila* larvae. 

Two classes of multidendritic neurons—class I [69,120] and bipolar multidendritic neurons [48]—have been implicated in proprioception. These two classes have been proposed to send segment-specific signals to the brain when muscle contraction in a segment is completed. This process serves to propagate the neural activity wave to the next segment. The proprioceptive neurons may also help to terminate contraction in a segment [69,85].

There are three types of class I multidendritic neurons in each body segment of larva: a single ventral neuron (Vpda) and two dorsal neurons (ddaE and ddaD) [119]. The neuronal dendritic arborizations of these neurons create a non-overlapping receptive field in the basolateral surface of epidermal cells [119,121]. During locomotion, as muscle segments contract, the dendrites of these sensory neurons get bent and deformed. The Vda neurons in the body segment provide synaptic inputs via GDL interneurons to A08e3 (Even-skipped positive interneuron) neurons that control left-right motor coordination [82]. Together, the A08e3 neurons and GDL interneurons integrate inputs from somatosensory neurons and regulate proprioceptive signals within larval segments, indicating the possible role of GDL interneurons in controlling the speed of locomotion and posture. 

During the muscle segment contraction cycle, the dendrites of ddaE and ddaD are recruited differentially in a direction-specific manner relative to the direction of movement. During forward locomotion, the ddaE dendrites are the first to be recruited and deformed; the ddaD dendrites are distorted later in the cycle. During backward locomotion, however, it is the ddaD dendrites that undergo bending, and the ddaE dendrites that bend later [119]. Interestingly, dendrites of class I multidendritic neurons show low responses to externally applied mechanical forces, suggesting that these neurons have specific roles in proprioception.

The bending or tilting of dendrites of class I multidendritic neurons causes compression, which might lead to an increase in dendritic force or tension, causing activation of proprioceptors [90]. Two mechanosensitive channels that serve as proprioceptors have been described: a transmembrane channel-like (*tmc*) and Piezo-like channels (*Pzl*) [122,123]. The *tmc* channels express in dendrites of class I multidendritic neurons [122,123] while *Pzl* are expressed in Chordotonal neurons. *tmc* mutants show defective and uncoordinated movements [122,123]. Similarly, *Drosophila* larvae with deleted *Pzl* gene showed characteristic locomotor crawling defects such as head lifting, backward movements, and lateral body bending [119]. Reduction of *Pzl* expression in Chordotonal neurons (iav-Gal4 > UAS-Pzl-RNAi) revealed similar phenotypes as *Pzl*-KO larvae, indicating that *Pzl* functions as a proprioceptor in Chordotonal neurons to regulate larval locomotion. Calcium imaging of Chordotonal neurons showed calcium responses upon 10 um displacement of the body wall, and these responses were abolished in *Pzl* mutants, suggesting that *Pzl* is required for ensuring Chordotonal neuronal sensitivity to body wall displacements. Together, these findings suggest the important role of the *tmc* and *Pzl* genes as proprioceptors and regulators of larval movements.

### 5.6. Phototaxis

During navigation, *Drosophila* larvae exhibit several photo-responsive behaviors, ranging from forward movement sequences, runs, turns, head-sweeps and pausing behavior. *Drosophila* larvae display negative phototaxis (moving away from light) during foraging behavior [124]. A sensorimotor circuitry map that mediates distinct light-guided behaviors has been identified, revealing that Bolwig’s organs (BO) and Md-IV are the main photosensory structures that respond to a range of light intensities and regulate phototaxis [125]. Two sets of photosensors mediate this light-guided phototaxis behavior: rhodopsin expressing in the BO and non-rhodopsin expressing Md-IV neurons.

These two photosensors differ in their responses to light intensities: BO neurons-based sensors respond to low light intensities (<0.08 W/m^2^) while Md-IV neurons respond to high light intensities (7–10 W/m^2^) [125]. The distinct responses to different light intensities is correlated to the functional behaviors they exhibit. The BO neurons mediate direction-based phototaxis and dark light induces pausing. The Md-IV neurons respond to noxious bright light and induce turning behavior [126], photosensing of MD-IV neurons relies on the expression of TrpA1 cation channels and the gustatory receptor 28b [101,127].

During phototaxis, spatial and temporal changes in light intensities affect the frequency of larval turns, and in response to different light gradients, larvae run towards a favorable light direction. Larvae use head sweeps to explore local luminosity gradient environments and upon identifying the favorable direction of the light (the darker side), run more (Figure 8A). BO are the main photosensory structures controlling these light-responsive behaviors. The BO is composed of 12 photoreceptors: four Rh5 and eight Rh6 types [127]. 

Photoreceptors required for light avoidance tile the body wall of *Drosophila* larvae. Under normal light conditions, both types of photoreceptors are required for navigation [128,129,130]. Meanwhile, during low-intensity light avoidance, Rh5-expressing, not Rh6-expressing photoreceptors are required [125,127,128,129]. The Rh6-expressing photoreceptors have been implicated in controlling temporal light sensing and navigation [129,130] and are required for sensing and avoiding green illumination (Figure 8B) [126]. 

The Rh5-expressing photoreceptors extend their axons to three types of target neuron; the optic lobe pioneer neurons, serotonergic neurons, and five lateral neurons (LNs) [127]. The five LNs are important for phototaxis behavior and are further divided into four pigment dispersing factor positive (PDF) cells and one PDF-negative cell (also known as the fifth LN neurons). The PDF-negative fifth LNs promote direction-based fast phototaxis behavior (Figure 8B). The fifth LN neurons are also involved in dark-light mediated pausing behavior [125,127,128,129]. 

In response to light or touch stimuli, *Drosophila* larvae may exhibit turning behavior, and might be involved in various aspects of escape, such as sensing vertical environment and turning away from the noxious stimuli [37,92]. Interestingly, by blocking rearing behavior, turning behavior is also blocked. Serotonergic neurons in the VNC suppress light-induced turning behaviors in *Drosophila* larvae (Figure 8B). These serotonergic neurons innervate the 5HT1B-receptor-bearing and leucokinin releasing neurons [37].

## 6. Sensory Feedback Neurons Are Critical for Motor Circuit Maturation

Sensory feedback neurons are mainly required to modulate motor behaviors. However, whether sensory feedback neurons are required during the maturation of the motor circuit is unclear. Fushiki et al. [131], reported that sensory inputs during embryonic development are required for motor circuit maturation. Eliminating the sensory inputs from sensory feedback neurons such as MD and Cho neurons during the late embryonic stages for 6 h had lasting effects on the sensorimotor circuit and affected larval crawling in mature second and third instar larvae. Inhibiting Cho but not MD neurons also had long-lasting effects on larval crawling, evidenced by decreased locomotion speed [131]. Calcium imaging of Cho neurons during larval peristalsis showed that Cho neuronal activity correlated with body wall contractions, also suggesting that Cho neurons help regulate the maturation of motor circuits. Interestingly, a developmental time window of 16 to 20 h after laying eggs is the critical time determined during embryonic development that is essential for the maturation of sensory motor circuits during larval development [131].

## 7. Conclusions and Future Directions

Larval locomotion or crawling is an elementary behavior and involves bilateral symmetrical body wall contractions. Motor neurons (MNs) innervate the body wall muscles in each body wall segment, while premotor interneurons project bilaterally to these MNs [26]. Ultimately, a neural circuit that generates the precise motor activity pattern comprises MNs, premotor interneurons, proprioceptors, and command-like descending neurons. Finally, neuromodulators such as dopamine or serotonin, can modulate motor output by reconfiguring neural properties such as excitability or synaptic efficacy and function [132,133]. The resulting coordinated activity in motor and premotor interneurons leads to rhythmic activity waves that manifest as motor output. With the continual advancements to the *Drosophila* genetic tool kit, as well as functional imaging, optogenetics, and connectomics we are now starting to rapidly characterize the role of neurons and neural circuits controlling normal and noxious stimulus-mediated locomotion in *Drosophila* larvae. Thus far, the neural circuits starting from premotor interneurons to MNs governing forward and backward locomotion of *Drosophila* larvae have been dissected in great detail. However, the neural circuits at the single neuron-muscle fiber level remain unclear. 

It is now emerging that specific neural pathways can be involved in more than one type of motion or sensory systems. For example, one study has implicated visual opsins in proprioception [134] and another implicated the gustatory role of chemoreceptors in wing mediated odor detection [135], indicating that multimodal cue-induced sensory behaviors may be regulated at multiple layers. Others have used refined intersectional genetic tools such as split-Gal4 [136] or zinc finger-based intersection methods [137], trans-synaptic labelling methods, activity depended GRASP and imaging methods for dissection of neural circuits [116,138,139,140,141], and fast optogenetic actuators such as crimson [18] and GtACR1 [17], to study the necessity and sufficiency of a single neuron to single muscle pathway. By these approaches, understanding which neural pathways control specific parts of behaviors may not be far away. In *Drosophila* larvae, nociceptive pathways have mainly been studied from neural circuitry perspective, with very few studies implicating genes in the functioning of these neural circuits. While connectomics, live imaging, and optogenetics provide information on connectivity, activity and behavioral relevance, molecular profiling of neurons at a single neuron offers a way forward to understand each neuron’s molecular level identity and their role in various neural circuits [142,143]. Learning from *Drosophila* serves as a primer for human studies, diseases, and therapeutics [144]. In addition, perhaps technological developments that can be readily tested in *Drosophila* might also be applied to other model systems. Understanding the neuroanatomy of *Drosophila* larva opens doors to understanding the neuroanatomy of other model organisms and humans.

## Figures and Tables

**Figure 1 biology-10-00090-f001:**
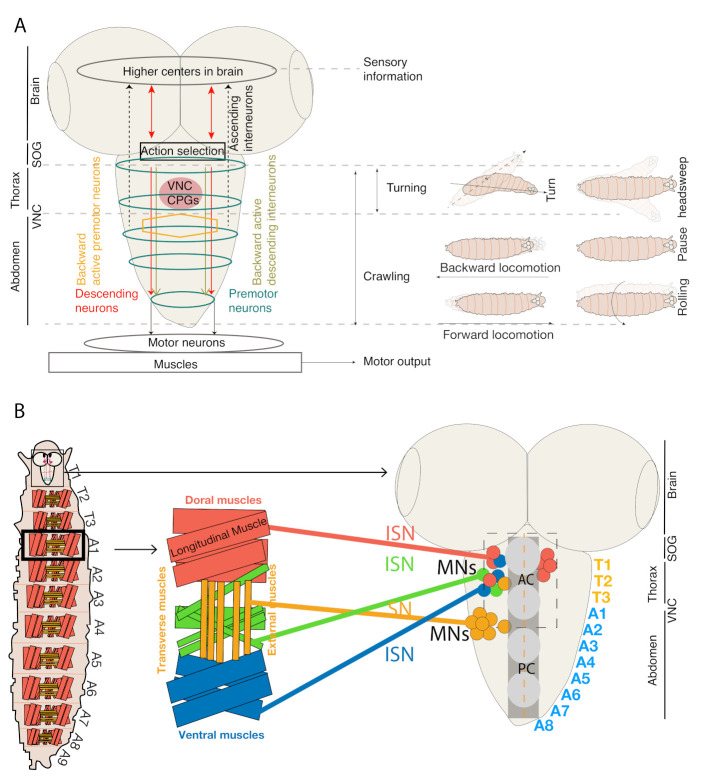
Schematics of topological organization, myotopic map and CNS regions involved in locomotion. (**A**) Schematics of organization of various types of neurons and thoracic and abdominal regions involved in various larval locomotor behaviors. (**B**) Schematic representation of *Drosophila* larval central nervous and neuromuscular system, which is divided into brain, subesophageal ganglia (SOG), and ventral nerve cord (VNC). In VNC, the positions of MNs cell bodies and representative muscle innervations are shown. MNs innervate abdominal hemisegments of each muscle through an intersegmental nerve (ISN, shown in red, blue and green), and segmental nerve (SN, shown in golden) which innervate internal and external (shown in golden) muscles, respectively. MN dendrite position in relation to their target muscles form a myotopic map. AC—Anterior commissure, PC—Posterior commissure (adapted and modified from [16]).

**Figure 2 biology-10-00090-f002:**
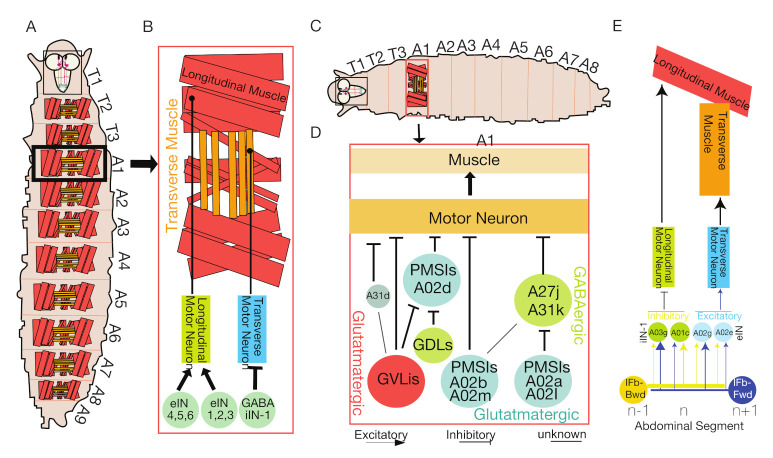
Interneuronal circuits regulating intrasegmental muscle contractions. (**A**) Schematic of *Drosophila* larva (not to scale) showing the anatomy of interneuronal circuits controlling coordinated longitudinal and transverse muscle segments. (**B**) Schematic showing neuromuscular connectivity (not to scale) of Longitudinal Muscles (LMs) (light red) and Transverse Muscles (TMs) (golden) that are innervated by longitudinal and transverse MNs, respectively (adapted and modified from [13,22]). (**C**) Schematic (not to scale) showing period positive median segmental interneurons (PMSIs) interneuronal circuits controlling MNs bursts duration. (**D**) PMSIs (A02b and A02m) provide inhibitory inputs to MNs. Other sets of PMSIs (A02a, A02I), presynaptic to GABAergic premotor neurons, remove inhibition from MNs after the relaxation of their target muscles. GVLIs inhibit MNs and control the termination of MNs bursting by being active at later phases of the locomotor cycle (adapted from [13,22]). (**E**) Intersegmental feedback neurons Ifb-Fwd and Ifb-Bwd neurons synapse to their target postsynaptic excitatory interneurons (A01c and A03g) that innervate transverse MNs and muscles and inhibitory interneurons (A02e, A02g and A14b) innervate longitudinal MNs and muscles for the regulation of bilateral motor coordination (adapted and modified from [22]).

**Figure 3 biology-10-00090-f003:**
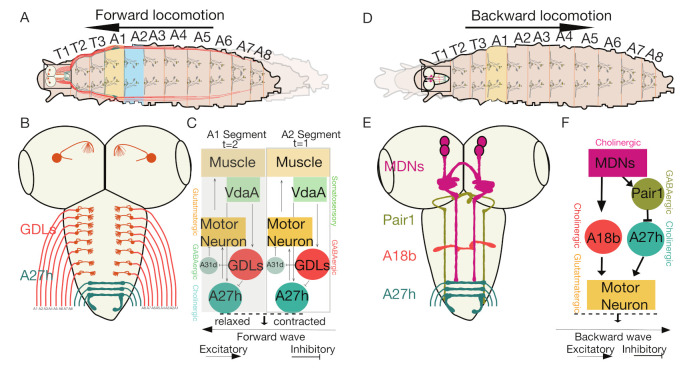
Neuronal pathways controlling forward and backward locomotion. (**A**) Schematic of *Drosophila* larvae (top view) depicting forward locomotion. (**B**) Schematic showing the anatomy of the larval CNS depicting expression of GDL interneurons (red) and A27h neurons (green) in VNC. (**C**) Neural circuit in a hemisegment of muscles leading to the generation of forward locomotion (adapted and modified from [25]). (**D**) Schematic of *Drosophila* larvae (top view) depicting backward locomotion. (**E**) Schematic showing anatomy of larval CNS depicting expression of descending MDN, inhibitory Pair1, interneurons A18b, and premotor A27h neurons in brain and VNC of *Drosophila* larvae [23]. (**F**) A neural circuit in a hemisegment of muscle eliciting backward locomotion (adapted and modified from [23]).

**Figure 4 biology-10-00090-f004:**
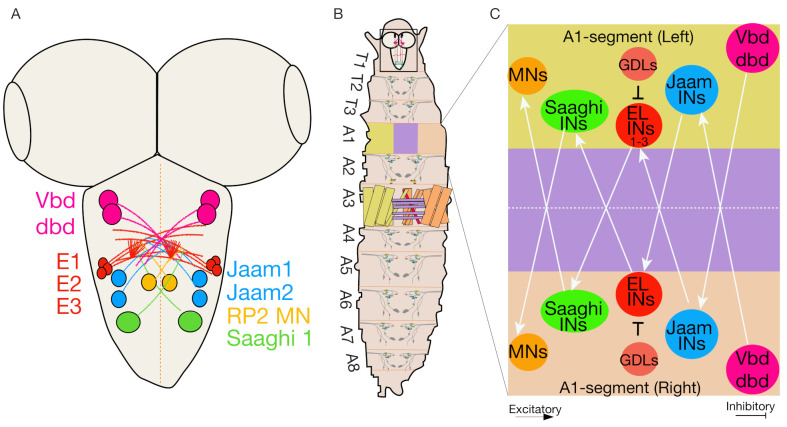
Neural circuits are involved in bilateral motor coordination. (**A**) A schematic showing the anatomical location of cell bodies of neurons involved in bilateral motor coordination (adapted and modified from [50]). (**B**) A schematic showing a top view of larval body segments. The red/blue/green territories in the abdominal segment A1 represent muscle groups in each segment. Red represents dorsal LMs, blue represents TMs, and green represents ventral LMs. Sensory nerves in A2 abdominal segments and MNs output to muscles (colored diamond, circles, and squares). (**C**) The EL (Eve positive lateral) interneuron sends bilateral connections to Jaam interneurons, Saaghi interneurons and MNs (RP2-output neurons), receives inputs from proprioceptive sensory neurons (Vbd and dbd) and controls bilateral motor coordination. The ELs also receive inhibitory input from GABAergic GDL interneurons (adapted and modified from [82]).

**Figure 5 biology-10-00090-f005:**
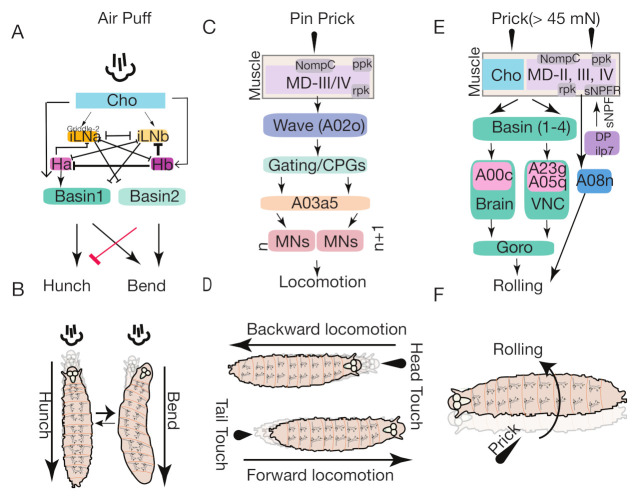
Neural circuit involved in mechanosensory-nociception. (**A**) The neural circuitry involved in promoting hunching and bending in response to a gentle air puff (adapted and modified from [30]). (**B**) Schematic depicting larval hunching and bending in response to an air puff. (**C**) The neural circuitry involved in location-specific, gentle touch-mediated forward and backward locomotion (adapted and modified from [24]). (**D**) Schematic depicting touch-mediated forward and backward locomotion. (**E**) The neural circuitry involved in noxious, touch-mediated rolling behavior. (**F**) Schematic showing larval rolling in response to a noxious mechanical touch (>45 mN [104]) (adapted and modified from [24,29]).

**Figure 6 biology-10-00090-f006:**
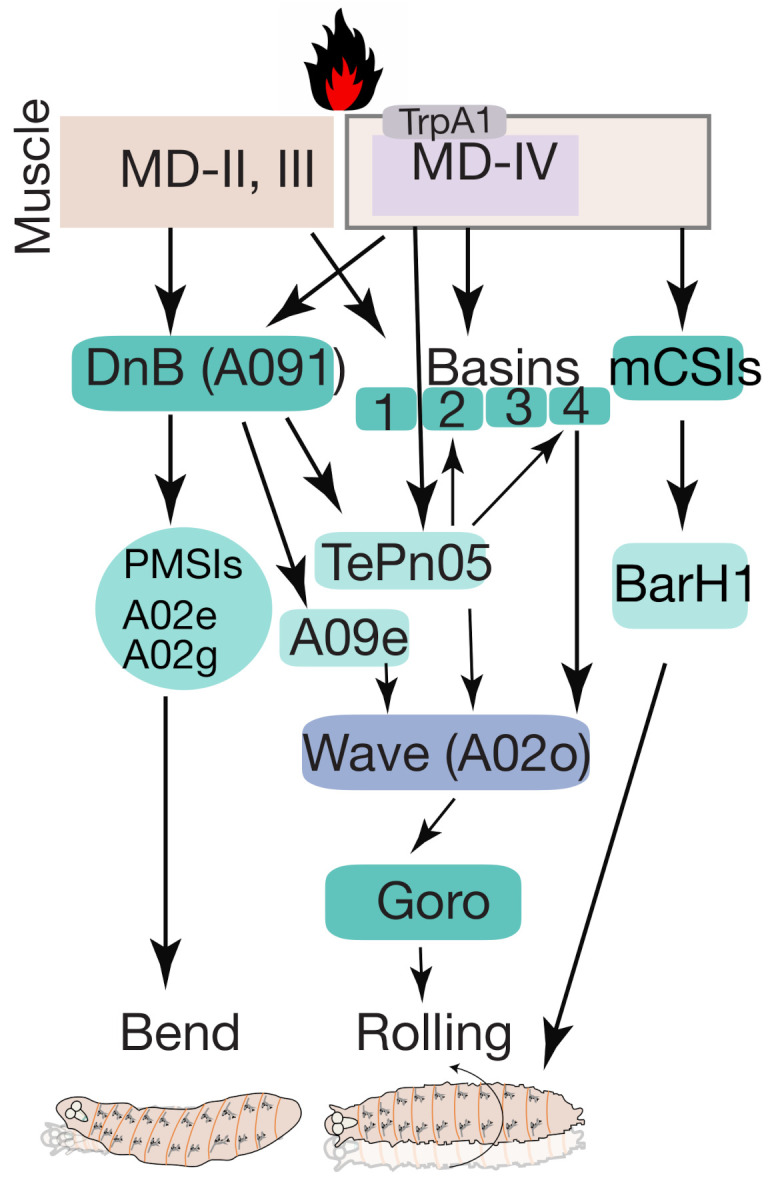
The neural circuits involved in thermo-nociception-mediated bending and rolling. Neural circuits depicting multiple pathways that can be recruited when larvae experience noxious heat (adapted and modified from [31,105]).

**Figure 7 biology-10-00090-f007:**
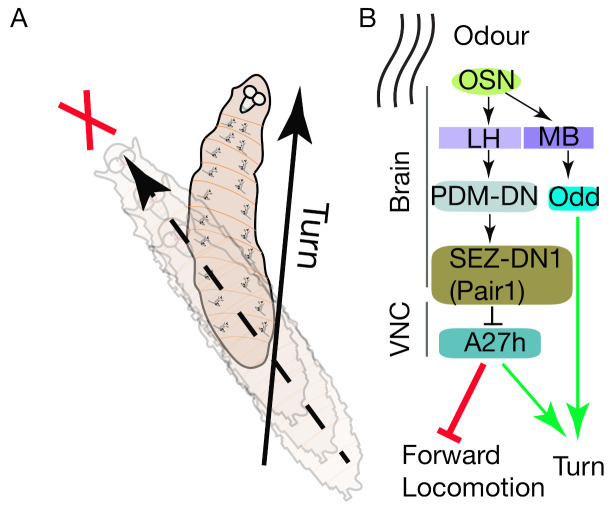
Neural circuit involved in chemotaxis. (**A**) Schematic depicting larval turning towards attractive food odor. (**B**) Neural circuits showing chemotactic pathways involving either MB or LH leading to inhibition of forward locomotion and promoting turn towards odor (adapted and modified from [32,80]).

**Figure 8 biology-10-00090-f008:**
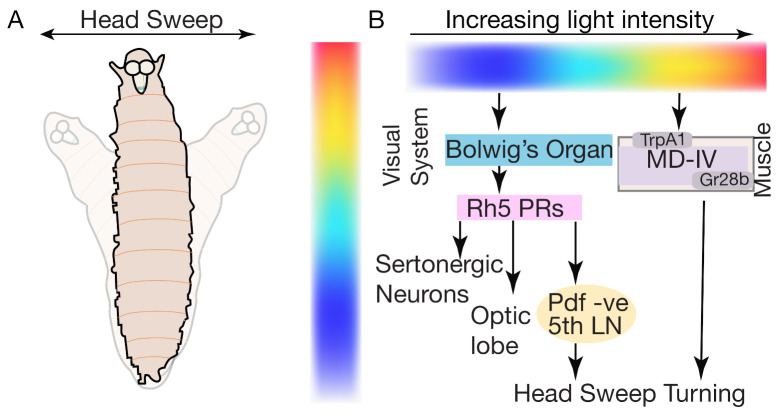
Schematics of neural circuit involved in phototaxis behavior. (**A**) Larval schematic showing head sweep behavior (**B**) Neural circuit showing phototaxis pathway, which mediate head sweep or turning behavior in response to low and intense light, respectively (adapted and modified from [125,127]).

**Table 1 biology-10-00090-t001:** List of neurons and associated drivers involved in forward and backward locomotion in *Drosophila* larvae. The name of the neurons, the associated Gal4 driver available from Bloomington *Drosophila* Stock Center (#BDSC), the type of segmental coordination, and the published function of these neurons are listed here.

Neurons	Neuron Identity	Drivers (#BDSC Stock)	Segmental Coordination	Function	References
GDL (A27j2)	GABAergic interneurons	GDL-Gal4 (9-20-Gal4, iav-GAL80),	Inter-segmental coordination	Forward and Backward locomotion	[25]
PMN (A27h)	Cholinergic premotor interneurons, excitatory	R36G02-Gal4 (#49939)	Inter-segmental coordination	Forward locomotion	[25]
PMN (A18b)	Cholinergic premotor interneurons, excitatory	R94E10-Gal4 (#40689) R94E10-LexA	Inter-segmental coordination	Backward locomotion, Escape behavior	[23]
Ifb-Fwd (A01d3)	Cholinergic feedback neurons	SS02065-Gal4 (R32C05-p65AD, R38E10- GDBD)	Inter-segmental coordination	Forward locomotion	[22]
Ifb-Bwd (A27K)	Cholinergic feedback neurons	SS026694-Gal4 (VT020818-AD R91E03-GDBD)	Inter-segmental coordination	Backward locomotion	[22]
Wave neurons (A02o)	Cholinergic and glutamatergic	VT25803-Gal4 (#v202269)	Inter-segmental coordination	Backward Locomotion, Escape behaviour	[24]
MDNS	Cholinergic descending neurons	MDNS- Gal4 (R49F02-AD; R53F07-DBD)	Inter-segmental coordination	Backward locomotion	[23]
iIN1	GABAergic inhibitory interneurons	R83H09-Gal4 (#41311) SS01411-Gal4	Intrasegmental phase difference	Forward locomotion	[27]
PMSIs (A02b and A02m)	Glutamatergic inhibitory interneurons	Per-Gal4	Intrasegmental coordination	Speed of locomotion	[38]
PDM-DN	Cholinergic	PDM-DN-Gal4	N/A	Chemotaxis	[32]
Odd neurons	Cholinergic	odd-Gal4	N/A	Chemotaxis	[80]
SEZ neurons		NP4820-Gal4	N/A	Chemotaxis	[81]
SEZ-Subset neurons		R23F01-Gal4	N/A	Chemotaxis	[81]
Pair1/SEZ- DN1	GABAergic	R75C02-lexA lexAop-myr:GFP (#54365)	Intrasegmental coordination	Backward locomotion/chemotaxis	[23]
ELs		EL-Gal4	Bilateral motor coordination	C-bends and Wavy body postures	[82]
Griddle-2 neurons	GABAergic	JRC-SS0918- split-Gal4	N/A	Hunch behaviour	[30]
fbLN-Hb	GABAergic	JRC-SS0888- split-Gal4	N/A	Bend behaviour	[30]
Vda/Vdc	Somatosensory neurons	tutl-GAL4	N/A		[25]
Late-born ELs		R11F02-Gal4 (#49828)	Bilateral motor coordination	C-bends and Wavy body postures	[82]
Brain and SOG region		BL-GAL4		Exploratory Behaviour	[40,41,55,68,69,70,71,72,73]

**Table 2 biology-10-00090-t002:** List of neurons and their drivers related to sensory locomotion in *Drosophila* Larvae. Name of neurons/genes, associated Gal4 driver available from Bloomington *Drosophila* Stock Center (#BDSC), class of sensory neuron, and their associated type of sensory behavior is listed here.

Neurons/Genes	Drivers (#BDSC Stock)	Class of Sensory Neurons Expressed	Thermo-Nociception	Chemo-Nociception	Mechano-Nociception	References
Md-1V sensory neurons	PPK-Gal4,R38A10-LexA (#54106) MD-IV Gal4	Md-1V-multidendritic neurons	+	+	+	[105]
Chordotonal neurons	Ch-Gal4	Mechanosensory	+		+	[24,29]
mSCIS	R94B10- Gal4 (#41325)	Md-1V-multidendritic neurons	+			[105]
Bar-H1 (SNa) motor neuron	BarH1- Gal4	Md-1V-multidendritic neurons	+			[105]
Basin Neurons	R72F11-Gal4 (#39786)	Md-1V-multidendritic neurons		+	+	[29]
Goro Neurons	R69F06-Gal4 and LexA	Md-1V-multidendritic neurons and class-II and Class-III sensory neurons			+	[29]
Down and back Neurons (DNB)	412 -Gal4	Md-1V-multidendritic neurons and class-II and Class-III sensory neurons	+			[31]
Foraging (*for*) gene	pr1-Gal4	Tracheal multidendritic neurons on larval body wall	+			[106]
A08n	R82E12 -Gal4 (#40183)	Md-1V-multidendritic neurons			+	[107]
Dp-ilp7	ilp7- Gal4	Md-1V-multidendritic neurons and class-II and Class-III sensory neurons			+	[107]
Basin-1	R20B01-Gal4 (#48877)	Md-1V-multidendritic neurons			+	[24,29]
Chordotonal neurons	iav-Gal4 (#52273)	Mechano-sensory	+		+	[24,29]

## Data Availability

Not applicable.

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
