# Peer review of "Anatomy and Neural Pathways Modulating Distinct Locomotor Behaviors in Drosophila Larva"

_biology, 2021, doi:10.3390/biology10020090_

Round 1

Reviewer 1 Report

This is a detailed and quite specialized review on the Drosophila neurobiology of locomotion. It is understandable, but the are many English and editing mistakes that make it difficult for me to read it. I kindly proof-read almost all the manuscript. I ask the authors to terminate the corrections appropriately following my instructions.

Below, I list some recurrent mistakes that I found:

  • There are many commas before “and” that should not be there. The comma before “and” is used only when more than 3 items.
  • There are many commas between subject and verb (a very serious error!).
  • In general, you need to decide between singular or plural when both are possible, but do it correctly. For example, at line 19, it is either “a successful movement” or “successful movements”. There are many other cases that need to be corrected or decided.
  • The article “the”, is always used when a singular noun is preceded by an adjective. Not if it is not and often not for plural nouns.
  • Adjectives are always singular.
  • “Which” is always preceded by a comma. “That” never.
  • The words of an acronym should be capitalized in the letters that make the acronym (lines 49 and 93 and so on).
  • I put question marks next to sentences or words that are not clear.
  • Genes should be capitalized as in FlyBase.
  • The “’s” genitive is used for people. Things become adjectives (ex.line 168).

As far as the content:

  • line 306: explain/define optogenetics (line 132) and optogenetic activation (line 306)..
  • line 590: ACV needs to be defined.
  • I think reference 82 (line 576) has been mis-interpreted. If I understood correctly, they manipulated one PDM-DN neuron. You write one and then plural. Decide.
  • line 580: A27h needs to be defined. Hours?
  • line 582: SEZ-DN1 one neuron or many?
  • line 589: reference 82 has nothing to do with odd. Instead, it would be good to cite a reference on odd expression.
  • lines 599-610: reference 84 should be explained better. 
  • The legends to the tables are missing.
  • All genes should be italicized.
  • The last sentence of the abstract about disorders, has not been developed. It should be removed or at least a paragraph about disorders should be added.

This is an interesting review that shows how Drosophila can be used to understand complex biological events that are shared by all animals. The next step will be to apply this knowledge for the study and the treatment of human malfunctions that may lead to disease. It is impossible for me to verify all the references. I only checked references 80-84 and asked some corrections (see above). A more specialized reviewer needs to evaluate the manuscript.

Finally, I suggest the authors to try to integrate in the review this article on olfaction:

  • Raad, H., Ferveur, J.-F., Ledger, N., Capovilla, M., & Robichon, A. (2016). Functional Gustatory Role of Chemoreceptors in Drosophila Wings. Cell Reports, 15(7), 1442–1454. http://doi.org/10.1016/j.celrep.2016.04.040.

Although it is about adults, it is a clear example of the power of Drosophila to understand the genetic control of a sensory function from which specific movements depend.

Reviewer 2 Report

In this paper, authors present a thorough review of the literature regarding the neural circuits underlying Drosophila larval locomotion. They covered most of the key studies on (1) premotor and motor circuits controlling coordinated muscle contractions underlying forward and backward locomotion of the larva and; (2) the neural circuits controlling sensorimotor transformations for different sensory modalities ranging from nociception to chemosensation. However, the review could be significantly improved by clarifying some concepts elaborating on the take-home messages from key papers and correcting some mistakes regarding the genetic reagents, neuronal cell types and their function. 

Concepts:

1- Central pattern generators (CPGs): Authors mention CPGs in multiple contexts in the paper. Although, the extensive work on electrophysiological and calcium imaging recordings of motor neurons from ex-vivo preps of the larval CNS suggest that CPGs underlie larval locomotion, the neural substrate of larval locomotor CPG is yet to be unraveled (if it exists at all). A section discussing the work by Stefan Pulver, Jimena Berni and Akinao Nose and speculating how the interneurons mentioned in the paper might integrate into the larval CPG would be useful.

2- Internal representations and motor neuron processing: In lines 48:50, the authors state that sensory system generates internal representation and passes this information to the motor neurons. Authors should clarify what a sensory system is and provide experimental evidence that they indeed generate internal representations. Since this is out of the scope of this paper, it would be useful if they refer the reader to other reviews on this subject. In addition, to my knowledge internal representations are unlikely to be passed onto the motor neurons directly bypassing descending/ascending/local interneurons and premotor circuits. I think the authors should conceptually elaborate on different levels of information processing underlying action selection from sensory neurons to high-order centers to descending neurons to premotor circuits. 

3- Topological organization of the ventral nerve cord and the body segments: Authors present detailed explanation of intrasegmental phasic relationships between longitudinal and transverse muscles to coordinate locomotion. They also present many neuronal cell types that are involved in intersegmental coordination underlying larval locomotion. To summarize the vast amount of information and put them in context, it would be clearer if they could first present the topological mapping between the neuropil segments in the VNC and the body segments of the larva. Then, they could present detailed explanation of (1) how symmetry of segmental neuronal activity in the VNC maps on sequential and symmetrical contraction of segmental muscles to mediate axial locomotion; (2) how asymmetrical activity in the anterior segments lead to head casting; (3) how asymmetrical contraction along multiple segment can result in turning behavior and finally; (4) how the different types of interneurons mediate these behaviors by acting either intra or intersegmentally.

4- Elementary modules of larval locomotion and multisensory integration: The authors should better explain the elementary behavioral modes of larval locomotion and underlying muscular and neural activity patterns (forward/backward locomotion, head casts, turns, rolling etc.), probably by using a graphical representation. This would further help the reader understand how probability of transitions between these elementary modes is regulated by the sensory input to generate adaptive locomotion, which would be of general interested in the field of motor control. Similarly, discussing how action selection is controlled by multisensory integration (e.g. Ohyama et al., 2015) to generate adaptive motor patterns would be of general interest.

Corrections/additions:

1- Line 19 and 20: ‘natural and defensive state’. Defensive state/behavior is also natural.

2- Line 46 and 47: Reference 5 is not appropriate for this sentence.

3- Line 59-61: Although I agree with the argument, the author should still refer to papers supporting it.

4- Line 67: Authors should refer to recent Drosophila connectome papers and milestone papers in the larva efficiently using the connectome.

5- Authors should cite GCaMP, trans-tango and GRASP papers.

6- Line 75 and 76: Authors should clarify the distinction between the body segments and muscles and the neuropil segments. Interneurons do not innervate muscle segments but neuropil segments.

7- Line 88-90: Authors should clarify that the not all behaviors stated in line 86 and 87 are due to symmetrical contractions in anterior-posterior axis. For example, head sweeps and turns are mediated by asymmetrical contractions.

8- Line 155: Motor neurons, interneurons and sensory neurons mainly covers all type of neurons in the larval nervous system.

9- Line 172: Citation is needed. Heckscher et al., 2012?

10- Table 1: Pair 1 and SEZ-DN1 are the same neuron labeled differently by different labs. This neuron inhibits generation of forward locomotion in the posterior segments to facilitate reorientation in chemotaxis (Tastekin et al., 2018) and backward locomotion (Carreira-Rosario et al., 2018).

11- Line 192 and 193: Interneurons do not innervate muscles.

12- Line 203-205: Reference needed.

13- Line 301: Mooncrawler descending neurons are annotated as MDN neurons in the original paper Carreira-Rosario et al., 2018. Authors could use the same annotation.

14- Line 304-306: 75C02 labels Pair 1/SEZ-DN1 neuron. It does not label A18b.

15- Line 313: Authors could cite the adult moonwalker neuron paper, Bidaye et al., 2014.

16- Line 478: Authors should elaborate on key findings of Ohyama etl al on how Basin circuit performs multisensory integration to adaptively control action selection.

17- Line 505: sNFP>sNPF

18- Line 572: Cite Gomez-Marin et al., 2011 and/or Gershow et al., 2012.

Round 2

Reviewer 1 Report

I see the authors made the corrections I asked, but although I did not re-read the whole manuscript I saw some mistakes I marked in yellow and in comments. What needs to be done, is to put all the genes in italics (example tic and Pzl at lines 812-825). Even when you write "tic channels", tmc is the gene and should be in italics and certainly when you write "Pzl functions".

Reviewer 2 Report

The revised manuscript is significantly improved compared to the first version in terms of content and clarity. The authors resolved the issues that were pointed out in the first round of revision. The current version represents a thorough review of Drosophila larval locomotion with detailed explanations of the key studies in this field.

However there are still some mistakes regarding the English language and style.

I recommend accepting the paper after resolving the language and style issues.